# Preparation, Characterization, and Anti-Cancer Activity of Nanostructured Lipid Carriers Containing Imatinib

**DOI:** 10.3390/pharmaceutics13071086

**Published:** 2021-07-16

**Authors:** Hafiz A. Makeen, Syam Mohan, Mohamed Ahmed Al-Kasim, Muhammad Hadi Sultan, Ahmed A. Albarraq, Rayan A. Ahmed, Hassan A. Alhazmi, M. Intakhab Alam

**Affiliations:** 1Department of Clinical Pharmacy, College of Pharmacy, Jazan University, Jazan 45142, Saudi Arabia; aalbarraq@jazanu.edu.sa; 2Substance Abuse and Toxicology Research Centre, Jazan University, Jazan 45142, Saudi Arabia; smohan@jazanu.edu.sa (S.M.); haalhazmi@jazanu.edu.sa (H.A.A.); 3Department of Pharmacology, College of Pharmacy, Jazan University, Jazan 45142, Saudi Arabia; malkasim@jazanu.edu.sa (M.A.A.-K.); rayanahmed@jazanu.edu.sa (R.A.A.); 4Department of Pharmaceutics, College of Pharmacy, Jazan University, Jazan 45142, Saudi Arabia; mhsultan@jazanu.edu.sa

**Keywords:** imatinib, NLC, Tween 80, SLS, MTT assay, release study, drug delivery, breast cancer

## Abstract

Breast cancer is the most widespread malignancy in women worldwide. Nanostructured lipid carriers (NLCs) have proven effective in the treatment of cancer. NLCs loaded with imatinib (IMA) (NANIMA) were prepared and evaluated for their in vitro efficacy in MCF-7 breast cancer cells. The hot homogenization method was used for the preparation of NANIMAs. An aqueous solution of surfactants (hot) was mixed with a molten mixture of stearic acid and sesame oil (hot) under homogenization. The prepared NANIMAs were characterized and evaluated for size, polydispersity index, zeta potential, encapsulation efficiency, release studies, stability studies, and MTT assay (cytotoxicity studies). The optimized NANIMAs revealed a particle size of 104.63 ± 9.55 d.nm, PdI of 0.227 ± 0.06, and EE of 99.79 ± 0.03. All of the NANIMAs revealed slow and sustained release behavior. The surfactants used in the preparation of the NANIMAs exhibited their effects on particle size, zeta potential, encapsulation efficiency, stability studies, and release studies. The cytotoxicity studies unveiled an 8.75 times increase in cytotoxicity for the optimized NANIMAs (IC_50_ = 6 µM) when compared to IMA alone (IC_50_ = 52.5 µM) on MCF-7 breast cancer cells. In the future, NLCs containing IMA will possibly be employed to cure breast cancer. A small amount of IMA loaded into the NLCs will be better than IMA alone for the treatment of breast cancer. Moreover, patients will likely exhibit less adverse effects than in the case of IMA alone. Consequently, NANIMAs could prove to be useful for effective breast cancer treatment.

## 1. Introduction

Breast cancer is the most prevalent malignancy in women worldwide and is curable in approximately 80% of patients with early-stage nonmetastatic disease. With the treatments presently available, advanced breast cancer with remote organ metastases is considered incurable [1]. On the molecular level, its progression is involved with the regulation of many pathways. Receptor tyrosine kinase overexpression is a typical pathway correlated with different malignancies [2]. At Novartis, imatinib (Figure 1) was synthesized and has emerged as the lead compound against chronic myelogenous leukemia (CML) cells for clinical development [3]. Centered on its application in the treatment of CML and gastrointestinal stromal tumors (GISTs), imatinib is a tyrosine kinase inhibitor that usually inhibits the tyrosine kinase action of BCR-Abl, platelet-derived growth factor receptors (PDGFRs), and KIT4 proteins. Tyrosine kinase receptors play an essential role in tumor development, pathologic angiogenesis, and the metastatic progression of breast cancer via the phosphorylation of target proteins [4]. It has been established that the inhibition of the function of the tyrosine kinase receptors involved in the EGFR signaling cascade forms the foundation for the usage of EGFR-specific inhibitors of tyrosine kinase, exemplified by gefitinib kinase [5]. Thus, the structurally similar drug imatinib is also expected to inhibit breast cancer. However, in cancer therapy, the therapeutic potency of imatinib mesylate is impaired by its off-target cardiotoxicity [6].

Lipid nanocarriers have great potential for the delivery of numerous anti-cancer agents. They have demonstrated benefits over traditional chemotherapy (e.g., drug resistance and adverse effects). The benefits associated with the particular properties of lipid nanocarriers include tumor-specific drug deposition, efficient pharmacokinetics and pharmacodynamics on account of the therapeutic agent, enhanced internalization and intracellular transport, and decreased biodistribution, resulting in the mitigation of the harmful consequences of anti-tumor treatments [7,8]. The administration of various drug molecules such as gefitinib [9], celecoxib [10], doxorubicin [11], duloxetine [12], tamoxifen [13], curcumin [14], and paclitaxel [15] has been extensively investigated using nanostructured lipid carriers (NLCs). Due to their fast intake by cells, biodegradability, and bio-acceptability [9], NLCs are being studied extensively for drug delivery. They have been shown to have better controlled release features and drug loading capability, as well as a lower burst release rate [16]. Furthermore, the use of biocompatible lipids and the involvement of a metabolic pathway in living systems decrease the possibility of toxicity [17]. NLCs are made up of two kinds of lipids: solid lipids (fat) and liquid lipids (oil). The inclusion of liquid lipids (oil) improves the drug delivery properties of NLCs compared to solid lipid nanoparticles (SLNs) [18]. As a result, NLCs are claimed to be a new generation of SLNs that integrate the qualities of SLNs while still overcoming the drawbacks.

The aim of this research was to design and formulate NLCs loaded with IMA (NANIMAs), then evaluate their in vitro characteristics such as particle size, polydispersity index, zeta potential, encapsulation performance, release performance, stability, and cytotoxicity (MTT assay) studies in MCF-7 breast cancer cells in order to minimize the therapeutic dose of IMA.

## 2. Materials and Methods

Imatinib was purchased from LC Laboratories (Woburn, MA, USA). Stearic acid (SA) and sodium lauryl sulphate (SLS) were obtained from Himedia (Mumbai, India). Sesame oil (pure) was obtained from a local merchant (Jazan, Saudi Arabia). Tween 80 (T80) was obtained from Loba Chemei (Mumbai, India). RPMI-1640 medium, FBS, and penicillin–streptomycin were purchased from Gibco (Invitrogen Corp., Waltham, MA, USA). Neutral red was purchased from Sigma Chemical Co. (St. Louis, MO, USA). All other chemicals and reagents used were of analytical grade.

### 2.1. Preparation of NANIMAs

NLCs loaded with IMA (NANIMAs) were prepared by the method of homogenization (hot). Stearic acid (fat) and sesame oil were used as lipid ingredients in the preparation of NANIMA formulations. The hot aqueous solution of surfactants (70 °C) in a beaker was transferred to another beaker containing a molten mixture of lipids and mixed with the help of a homogenizer (HG-15D, WiseTis, Germany). The mixing was carried out for a period of 20 min at a speed of 5000 rpm. The aqueous solution of surfactants was prepared by dissolving sodium lauryl sulphate (SLS) and Tween 80 (T80) in 25 mL of purified water (Table 1). The lipid phase consisted of stearic acid (SA) (0.5 g), sesame oil (0.25 mL), and IMA (0.05 g). The preliminary studies were conducted and the ratio of SLS and T80 was selected based on particle size and stability.

### 2.2. Particle Size, Polydispersity Index, and Zeta Potential

Photon correlation spectroscopy (Nano ZS90, Malvern, WR14 1XZ, UK) was used for the assessment and characterization of the parameters, including particle size (as mean), polydispersity index (PdI), and zeta potential. The study was performed after the proper dilution of formulations with Millipore water (1:30 *v*/*v*). The appropriate concentration of particles was obtained after dilution to circumvent multi-scattering actions. All measurements were made in triplicate.

### 2.3. Encapsulation Efficiency (EE)

The encapsulation efficiency (EE) was determined in terms of the percentage quantity of IMA trapped in the NLCs. The ultrafiltration centrifugation method was followed for the estimation of the EE (%) of all the NANIMAs. This method consists of the centrifugation of an ultra-filter tube (3 K, Millipore Ireland) containing 2 mL (≈4 mg of IMA) of the formulation at a speed of 4000× *g* for a period of 20 min. The free IMA was estimated in the filtrate obtained after centrifugation. The filtrate was analyzed spectrophotometrically after proper dilution. The spectrophotometric analysis was carried out at 265 nm by means of a spectrophotometer (UV, Shimadzu, Kyoto, Japan). The EE was estimated using the formula mentioned below:
$$EE\;\left(\%\right)\;=\;\frac{\left(Quantity\;of\;IMA\;taken-Quantity\;of\;IMA\;obtained\right)}{Quantity\;of\;IMA\;taken}\times 100$$

### 2.4. Lyophilization

The aqueous solution of mannitol (6% *w*/*v*) was prepared and used as a cryoprotectant during freeze drying of the NANIMA formulations. The aqueous dispersion of NANIMAs was mixed with a mannitol solution (1:4), followed by freezing in an ultra-low temperature freezer (U9280-0002, New Brunswick Scientific, Edison, NJ, USA) at −70 °C for 12 h. The frozen samples were lyophilized using a lab freeze dryer (BT85, Millrock technology, Kingston, NY, USA). After 24 h, the dried samples were collected and stored in suitable containers.

### 2.5. Scanning Electron Microscopy (SEM)

A scanning electron microscope (Zeiss EVO LS10, Carl Zeiss NTS, Oberkochen, Germany) was used for taking micrographs of the NANIMAs. Double-sided adhesive carbon tape (SPI Supplies, West Chester, PA, USA) was used for fixing the NANIMA samples. Moreover, the coating was carried out in an argon atmosphere at 20 mA for 60 s under a vacuum with gold in a Q150R sputter coater unit (Quorum Technologies Ltd., East Sussex, UK).

### 2.6. In Vitro Release Study

The in vitro release study was performed for all NANIMA formulations. The dialysis bag (12–14 KD MWCO, Spectrum Laboratories, Inc., Rancho Dominguez, CA, USA) method was used to perform the study. The formulation (1 mL; equivalent to 2 mg of IMA) was transferred to the dialysis bag, followed by fastening both ends of the bag with thread. The bag was placed in a beaker containing a phosphate buffer solution (pH 7.4; 300 mL) mixed with Tween 20 (1% *v*/*v*) as the release media. Stirring of the media was performed using a bead on a magnetic stirrer at a speed of 50 rpm and at a temperature of 37 °C. An aliquot (2 mL) was taken from the beaker after 0.5, 1, 2, 3, 4, 5, 6, 12, and 24 h, followed by replacement with fresh media. The collected sample was passed through a syringe fitted with a filter (0.45 µm), and a filtered sample was obtained. This was followed by analysis for drug content using a UV spectrophotometer (Schimadzu, Kyoto, Japan) at a λ_max_ of 265 nm. The wavelength was selected after scanning the solution containing IMA. The maximum absorbance at a wavelength of 265 nm was obtained.

### 2.7. Stability Studies

The NANIMA formulations were subjected to stability studies and analyzed under the influence of definite temperature. The NANIMA formulations were stored at room temperature (25 °C) for a period of three months. These were then analyzed for different physical changes, including consistency, color, aggregation, and smell. Moreover, the characterization parameters, including size, zeta potential, polydispersity index, and leakage of drug, were used to evaluate the influence of temperature.

### 2.8. Cell Line and Cell Cultures

The current study was conducted using MCF-7 cells (ER-positive breast cancer cells), which were purchased from ATCC. The cells were grown in a CO_2_ incubator (New Brunswick Scientific, Edison, NJ, USA) at 37 °C in using RPMI-1640 media, which was supplemented with 10% fetal bovine serum, penicillin (100 U/mL), and streptomycin (100 g/mL). The samples were diluted to the desired concentration using serum-free media and treated for the required time period.

### 2.9. Cell Viability Assay

The cytotoxicity assay was carried out by a colorimetric MTT viability assay [19]. Briefly, different sample concentrations were put in triplicate and incubated for 72 h. Each plate was included with untreated cell controls and a blank cell-free control. MTT (5 mg/mL) was provided to each well after incubation and the plates were further incubated for an additional 4 h, after which, the media were removed. To solubilize the formazan crystals, DMSO (100 μL) was inserted into each well. The absorbance was read using a microtiter plate reader at a wavelength of 490 nm (BioTek Instruments, Winooski, VT, USA). With the necessary controls taken into account, the percentage of cellular viability was determined. In triplicate, the experiment was completed. The cell proliferation inhibitory rate was estimated using the following formula:$$Growth\;inhibition\;=\;\frac{\left(OD\;control-OD\;treated\right)}{OD\;control}\times 100$$

The cytotoxicity of our formulations on cancer cells is expressed as IC_50_ values (the sample concentration reducing the cell count of treated cells by 50% with respect to untreated cells).

### 2.10. Detection of Apoptosis by Acridine Orange (AO) Propidium Iodide (PI) Double Staining

IMA- and S2TIN-induced cell death in MCF-7 breast cancer cells was quantified using AO and PI double-staining, according to standard procedures, examined under a fluorescence microscope. Briefly, treatment was carried out in T-flasks. Cells were plated at a concentration of 1 × 10^6^ cells/mL for 24 h, the media were substituted with new media containing 52.5 µM of IMA and 6 µM of S2TIN, along with a blank control. The cells were incubated for 72 h. Floating and adhering cells were collected, centrifuged at 300× *g* for 10 min, and washed twice with PBS to remove the remaining media. The supernatant was discarded, and the cells were stained with fluorescent dyes containing AO (10 mg/mL) and PI (10 mg/mL). The freshly stained cell suspension was dropped onto a glass slide and covered with a coverslip. The slides were observed under an ultraviolet fluorescence microscope within 30 min before the fluorescent color started to fade. The viable, early dependent phospholipid-binding protein, the plasma membrane alterations, and apoptotic and late apoptotic cells were observed.

### 2.11. Statistical Analysis

All data are reported as means ± standard deviations (SDs). For statistical analysis, a one-way ANOVA was used, and statistically significant differences are described as *p*-values less than 0.05 (*p* < 0.05).

## 3. Results and Discussion

The NANIMAs were prepared using fat and oil by the hot homogenization method [9]. Moreover, surfactants, including SLS and T80, were used for the stabilization of the system. The preliminary studies were conducted and the different ingredients, including surfactants, lipid phase, and their ratio, were selected based on solubility, stability, availability, and particle size. SA is a widely used ingredient present in the list of chemicals that are recognized as safe (GRAS) by the FDA [20]. It has an excellent stabilizing feature. Sesame oil was selected for its sustained release behavior, stability (does not become readily rancid) [21], and solubility of IMA. SLS (anionic surfactant) (excellent dispersion properties in homogenization process) and T80 (nonionic surfactant) are employed in a wide range of pharmaceutical formulations as emulsifiers and are easily available. The combined use of SLS and T80 gives particles of high stability [22]. A number of NANIMAs were prepared using various ratios of their ingredients and their stability was assessed. More stability and lesser particle size of the compositions was preferred for the study. The NANIMA formulations were evaluated for particle size, zeta potential, SEM, release studies, stability studies, and cytotoxicity studies.

### 3.1. Particle Size, Polydispersity Index, and Zeta Potential

The NANIMAs were evaluated with the purpose of determining their particle size distribution, zeta potential, and PdI values (Table 2). The influence of the surfactants (type and amount) on the particle size of the NANIMAs was analyzed. The size of the NANIMAs was observed to be greater (S2TIN > STIN) when the surfactant (SLS) concentration increased. However, contrasting results were observed in the case of T80 (size decreased after increasing the T80 concentration) (TSIN > T2SIN). However, the result was found to be insignificant. Thus, the particle size was insignificantly affected due to variation in the amount of surfactants (T80 and SLS) in the NLCs containing IMA.

Particle size is considered to be a fundamental factor in the development of nanotherapeutic systems. The composition of a formulation, including the surfactant, lipids, and drug, greatly affect the particle size. Moreover, the process parameters, such as the speed and duration of homogenization and temperature, positively affect the size of NLCs.

Particle size exerts its effect not only on stability, but also significantly affects the cellular uptake by tumor cells and the passive targeting of anti-cancer agents to tumors [23]. For the chemotherapeutic agents to be delivered effectively, the particle size are considered to be in the range of 50–200 nm. The appropriateness of the nanocarriers between 50 and 100 nm for avoiding monocellular phagocytic system uptake and prolonging blood circulation time was investigated. Nanocarriers smaller than 200 nm have been found to be effective at passively targeting tumor tissues through enhanced permeability and retention (EPR) [24].

The polydispersity index (PdI) of the NANIMAs is shown in Table 2. The PdI is reported to be a measure of the aggregation/agglomeration of particles in nanosystems. Thus, the monodispersed conduct of a system is predicted when the PdI value is closer to zero. Moreover, the polydispersity of the system is expected with the higher PdI value (>0.5). Monodispersed systems have a lower agglomeration propensity than polydispersed systems [25]. The physical stability of colloidal dispersions can be predicted by zeta potential (ZP) values, which are known as a primary predictor of nanocarrier stability, including NANIMA preparations. Furthermore, a certain ZP value improves a nanocarrier’s ability to bind to cell membranes. For the transport of therapeutic agents, a particular ZP value is needed to produce electrostatic contact with cell membranes [26]. A higher possibility of colloidal system stabilization can be achieved by the higher ZP values, irrespective of the type of charge (positive or negative). Characteristically, stability (no aggregation/flocculation) of the system is achieved when charged particles with a ZP value of (>|20|) are present [27]. Thus, this high a ZP value is appropriate for retaining electrostatic repulsion among similarly charged particles for the prevention of aggregation. The ZP value of a NANIMA is characterized as the potential contrast between the NLC surface and its fluid medium (i.e., the particle–liquid interface). The charge of all NANIMAs was found to be negative (negative ZP). The negative charge of NANIMAs may be contributed by slightly ionized fatty acids [28]. The ZP of the NANIMAs increased when increasing the quantity of T80. The highest ZP (negative) was observed for T2SIN, containing the highest amount of T80 among all of the NANIMAs. Thus, T2SIN is assumed to be most electrostatically stable among all of the NANIMAs based on its ZP value.

The lowest ZP value of STIN can be explained by the shielding of electrostatic charges on the NANIMAs by nonspecifically adsorbed nonionic surfactants (T80) [29]. Furthermore, the higher ZP value of S2TIN compared to TSIN can be explained by the availability of adsorption space on the surface of the NANIMA particles. Subsequently, the sequence of stable NANIMA formulations can be arranged as T2SIN > S2TIN > TSIN > STIN.

### 3.2. Encapsulation Efficiency (EE)

EE estimation was employed to differentiate the amount of IMA incorporated into the NLCs from the free IMA. The NANIMA formulations were found to exhibit high EE values (from 98.50 ± 2.43% to 99.79 ± 0.03%) (Table 2). The solubility of the IMA in the lipid phase accounted for the high EE value. Moreover, the high EE value may also be contributed by the partitioning of IMA between the aqueous and oil phases.

The accommodation of greater quantities of IMA in the NANIMAs (thus resulting in a high EE value) may also be explained on the basis of the introduction of sesame oil into stearic acid, which increases the imperfections in the crystal lattice and decreases crystalline order. It has been demonstrated that a lower degree of crystalline order in lipid nanocarriers is considered responsible for the increase in the capability of accommodating more drugs [30]. Hence, amid the fatty acid chains and/or lipid layers, the IMA molecules can be accommodated in higher amounts. Moreover, parallel outcomes were achieved previously [9,31,32,33,34].

The high EE value may also be attributed to the concentration of surfactants. The increase in the viscosity of the aqueous phase and the addition of surfactants may have resulted in the high EE value after decreasing the diffusion speed of IMA. The smaller size of the particles increased the surface area of the NANIMAs, where more IMA molecules were incorporated. Thus, the availability of adequate surfactant enabled the IMA to remain entrapped within the lipid, consequently leading to a high EE value. Furthermore, micelle-forming surfactants, including SLS, form monomers and micelles on the lipid surface that provide alternative sites for IMA incorporation [35]. The effect of the concentration of surfactant on the EE value was found to be in accordance with the outcomes of others [31,33,36].

### 3.3. Scanning Electron Microscopy (SEM)

SEM analysis was conducted to acquire information about the morphology (shape and size) of the NANIMAs. Two types of samples were used for this analysis: the first type was lyophilized samples, while the second type sample was prepared by dispersing lyophilized NANIMAs in water and drop-casting them onto a glass slide. The morphology and size of all of the NANIMAs were found to be in the nanometer range. However, the lyophilized NANIMAs revealed elongated and parallelepiped shapes (Figure 2). Thus, the NANIMAs with mannitol as a cryoprotectant seemed to adopt an orthorhombic form with flattened structures. Moreover, the agglomeration of particles was observed, which might be either be due to the nature of the lipid, the concentration of the dispersion medium, or the shrinkage of NLCs during the drying of the NANIMAs. The SEM images obtained after reconstitution revealed spherical shapes and more or less smooth surfaces (Figure 3).

### 3.4. In Vitro Release Study

Figure 3 depicts the release characteristics of the NANIMA formulations. The in vitro release tests of the NANIMA formulations were carried out using the dialysis bag diffusion technique. To avoid interference from IMA solubility in the release medium, the sink conditions were preserved throughout the study. The slow and sustained release of IMA was exhibited by all of the NANIMA formulations, as expected for NLC formulations. The release of IMA from within the NLC cores, as well as partitioning between the water and lipid matrix, may be attributing factors for the long-term release behavior of lipid nanocarriers such as NLCs [37]. It could also be attributed to the interfacial membrane’s barrier function, the solid matrix of NLCs, or the subsequent immobilization [27]. The slow release of the IMA from the NLCs could be explained by entrapment homogeneity throughout the system [28]. Furthermore, the release characteristics of the NANIMA formulations were supported the high EE value (>99%). Consequently, the prolonged anti-cancer action may be obtained from a single dose of NLCs containing an anti-cancer agent. Thus, the suitability of NLCs may be considered for the delivery of IMA. 

The release studies exhibited an absence of a burst effect in the release pattern shown in Figure 4. This may be attributed to the high EE value and the negligible amount of unentrapped IMA. Generally, the presence of unentrapped drug on the surface of NLCs may contribute to the burst effect in release studies [32,38]. As shown in Table 2, all of the NANIMA formulations were found to have an EE value of nearly 100% (>99%), which was supported by release tests. 

The IMA solution was prepared by dissolving IMA in ethanol and adding aqueous media. The release of IMA from the IMA solution was found to be faster than from the NANIMA formulations. This may be because of sustaining the release of IMA by the dialysis membrane only.

However, the NANIMA formulations revealed sustained release performance that may be credited to both NLC and the dialysis membrane. S2TIN released the highest and TSIN the lowest amount of IMA among all of the NANIMA formulations after 24 h of study. STIN and T2SIN released nearly equal amounts of IMA after 24 h. 

Thus, different factors may be considered responsible for affecting the release of IMA from NANIMA formulations, including the nature and amount of surfactants and the duration of sampling. SLS appears to be extra influential, compared to T80, on the release of IMA from NANIMA formulations (S2TIN > T2SIN and S2TIN > TSIN). Moreover, the higher amount of SLS provides a higher amount of IMA to be released (S2TIN > STIN). However, in the case of T80, an even higher amount (T2SIN) seems to be ineffective in mobilizing IMA excellently from the solid core of NLCs to be exhibited as the released amount more than the lesser amount (S2TIN) of SLS. Reliably, the collective impact of SLS and T80 on the release behavior of IMA was more recognizable in S2TIN than in STIN, TSIN, and T2SIN. This may be explained on the basis of the sampling gap and combined effect of the surfactants. A sampling gap of 12 h provided ample duration of time for the action of the surfactants, and accretion of the released IMA may be considered a dependable factor for a greater release. 

Moreover, the NANIMA formulations were evaluated for the mechanism of release of IMA. The release data of the NANIMAs were fitted into different kinetic models (Figure 5). The preferred model that best fit the release data was chosen on the premise of the higher correlation coefficient (R^2^) values. Accordingly, the Higuchi model (diffusion-controlled profile) was followed for all of the NANIMA formulations, including STIN (R^2^ = 0.974), S2TIN (R^2^ = 0.904), TSIN (R^2^ = 0.996), and T2SIN (R^2^ = 0.975). Consequently, all of the NANIMA formulations exhibited the release of IMA by the diffusion-controlled mechanism based on the Fick’s law, which is square root time-dependent.

### 3.5. Stability Studies

Among the factors affecting the stability of lipid nanocarriers, including NLCs, temperature is considered to be a significant one. Storage at room temperature (25 °C) does not require any special conditions. Thus, stability at room temperature (25 °C) is considered to be commendable. Accordingly, the NANIMA formulations exhibited a temperature effect on particle size, the polydispersity index (PdI), encapsulation efficiency (EE %), and zeta potential (ZP) after three months of storage. The outcome of the temperature effects was observed, as shown in Appendix A. The effect was observed in the favor of particle growth in the case of TSIN, followed by T2SIN, S2TIN, and STIN (i.e., TSIN > T2SIN > S2TIN > STIN). However, the particle size of the NANIMA formulations remained in the colloidal nanometer range (<550 nm) [39]. Since the increase in the size of TSIN was not more than two-fold, this indicates the nonappearance of aggregation. This kind of variation may possibly be ascribed to the swelling or adsorption of extra surfactants on the particle surface [18].

The disparity of the PdI values of all of the NANIMAs after three months of storage was compared and estimated with respect to zero months. The outcome of the effect of temperature on PdI is shown in Appendix A. The physical stability of nanoformulations may also be explained by the disparity of PdI values. The long-term stability of nanosuspensions is favored by smaller PdI values [40]. All NANIMA formulations exhibited an increase in their PdI values. Among these, STIN (*p* = 0.065) was found to exhibit the highest and T2SIN (*p* < 0.05) exhibited the lowest increase in PdI value with respect to zero months. However, S2TIN was found to reveal the lowest PdI value (zero months), but its increase in PdI was found to be more than that of T2SIN after three months of storage. Thus, on the basis of this perception, T2SIN may be considered the most stable in terms of long-term stability, followed by TSIN, S2TIN, and STIN (i.e., T2SIN > TSIN > S2TIN > STIN). Hence, T80 was found to provide better long-term stability to the NANIMA formulations than SLS. 

The results of the influence of temperature on the leakage of drug are shown in Appendix A. The leakage of IMA from NANIMA formulations was assessed by comparing EE (%) before and after the storage period. S2TIN was found to have the maximum EE value, while T2SIN exhibited the minimum value among all of the NANIMA formulations after three months. However, the NANIMA formulations were found to have a decreased EE value (with respect to zero months) when analyzed after the storage period. The rate of decrease in the EE value (leakage of IMA) was faster in T2SIN (*p* < 0.05), followed by TSIN (*p* < 0.05), S2TIN (*p* < 0.05), and STIN (*p* > 0.05). A commendable capability of preventing the leakage of IMA was exhibited by STIN over the other NANIMA formulations at room temperature (25 °C). Thus, SLS seems to be more effective than T80 in preventing the leakage of IMA. 

The effect of temperature on ZP is shown in Appendix A. The ZP values of all of the NANIMA formulations decreased when assessed at the end of the storage period. The newly formulated NLC-dispersions show high ZP values, decreasing in value with the increment of time passing. The maximum ZP was estimated for T2SIN, followed by S2TIN, TSIN, and STIN after the end of the storage period. However, the rate of reduction in the ZP values was faster in the case of STIN, followed by T2SIN, S2TIN, and TSIN (i.e., STIN > T2SIN > S2TIN > TSIN). The reduction in the ZP value was found to be insignificant (*p* > 0.05) for S2TIN, T2SIN, and TSIN. Thus, the electrostatic layer was retained everywhere around the NLC particles throughout the system. Consequently, these formulations may be considered to have great physical stability after storage for three months. Furthermore, STIN exhibited a significant reduction in ZP (*p* < 0.05). Several factors are responsible for affecting the ZP values, including the type and amount surfactants, variety in particle size and the property of the particles to agglomerate throughout storage. Thus, the charges and the dispersion force between particles decreased, which may be because of the variety in particle size (as indicated by the PdI) and the agglomeration tendency of the said particles [41].

### 3.6. Anti-Cancer Evaluation

A cytotoxicity assay (MTT) was conducted to determine the anti-cancer prospective of the different formulations, i.e., STIN, S2TIN, TSIN, and T2SIN, along with IMA alone (Figure 6). The results show that the S2TIN formulation achieved a significantly higher cytotoxicity, which was superior compared to the pure compound IMA. Moreover, the blank exhibited no cytotoxicity within the range used in this study (Table 3).

Furthermore, under a fluorescence microscope, the apoptotic, necrotic and viable MCF-7 breast cancer cells were scored. This also included the untreated control cells (200 cells) that were counted randomly and differentially. The assay showed that IMA (52.5 ± 3.24 µM) and S2TIN (6 ± 0.2 µM) displayed an equal number of apoptosis cells. There was a time-dependent increase in dead cells, as observed at 48 and 72 h (Figure 7A–E). Early apoptosis was apparent within the fragmented DNA by intercalated AO. The fluorescent bright green color could only be found in the treated cells. In comparison, it was found that the untreated cells had a green, intact nuclear structure. Differential scoring of the treated cells (200 cell population), as shown in Figure 7F, highlighted that the difference in positive apoptotic cells was statistically significant for 48 and 72 h (*p* < 0.05). On the contrary, as seen in Figure 7F, there was no statistically significant difference (*p* > 0.05) among the IMA and S2TIN formulations in various periods during the therapy (48 and 72 h).

In clinical use, cytotoxic medications have various functional challenges, such as a higher toxicity, low cancer cell specificity, and susceptibility to pharmaceutical formulations. In the case of IMA, cardiotoxicity is the primary concern [42]. Hence, reducing the effective dose will be an achievement in the treatment strategies. Our results obtained from the cytotoxicity assays revealed that the S2TIN formulation has more significant potential compared to IMA alone. Additionally, we checked the possibility of S2TIN to exert apoptosis in different doses. The phase of programmed cell death or apoptosis is typically distinguished by distinct morphological features and energy-dependent biochemical pathways [43]. Imatinib has been previously shown to induce apoptosis in breast cancer in vitro [44]. Our studies revealed that the interacting dye that binds with DNA is visible evidence of the induction of early and late apoptosis in the given doses. Moreover, the quantification of the cell populations in different phases suggested that S2TIN has the potential to induce apoptosis equivalent to IMA at lower doses.

The cytotoxicity of blank formulations was studied and no significant cytotoxicity was observed at the concentration used in the NANIMA formulations. SA is widely used in pharmaceutical formulations and is generally regarded as non-toxic and non-irritant material [21,45]. Sesame oil has been found to be cytoprotectant and non-cytotoxic [46]. T80 has been assessed to be non-cytotoxic and has potential as a simple, rapid and non-toxic method for delivering drugs to cells in culture [47]. The nano-drug delivery systems containing SLS revealed no toxicity to human cells on MTT assay method of cytotoxicity studies [22,48].

## 4. Conclusions

NANIMA (NLCs containing imatinib) formulations were successfully prepared and exhibited commendable outcomes for characterization parameters and evaluation studies. Different ingredients, including stearic acid, sesame oil, and surfactants (SLS and T80), were used in the preparation. Due to their smaller size (~100 nm) and lipid nature, NANIMAs ensure the adequate penetration of IMA over different membranous obstacles that will possibly bring about acceptable therapeutic outcomes. Moreover, the release studies ensure a therapeutic effect on the target site for an extended period. Furthermore, the high EE value (>95%) is an encouraging aspect for improved remedial effectiveness, as revealed by cell line studies. The Higuchi model describes the release of IMA from all NANIMA formulations. The effect of storage was observed not only on size, but also on PdI, EE, and ZP. However, the similarity of results may be attributed to the fact that the comparable conditions were provided for the preparation of all NANIMAs except the amount of surfactants. Thus, the same amount of lipids, water and IMA was used in all cases. Moreover, the process parameters, including the speed and duration of homogenization, remained same for all NANIMAs.

The in vitro anti-cancer potential of NANIMAs was evaluated after performing cytotoxicity studies on MCF7 cells. The anti-cancer effect of IMA-loaded NLCs was found to be 8.75 times (S2TIN) superior to IMA alone. Accordingly, a reduced dose will provide an enhanced anti-cancer effect on cancer patients. Consequently, less adverse effects and exposure of the body will be observed. Thus, S2TIN took the credit of being the optimized composition among all NANIMAs based on release and cytotoxicity studies. The commendable outcome observed by S2TIN may be credited to the mutual effect of not only its small size, lipid nature, and entrapment efficiency, but also the type of ingredients selected, optimized quantities of ingredients, and the method used. Therefore, the S2TIN NANIMA formulation may be considered suitable as an active cure of cancer after carrying out studies on animal models.

## Figures and Tables

**Figure 1 pharmaceutics-13-01086-f001:**
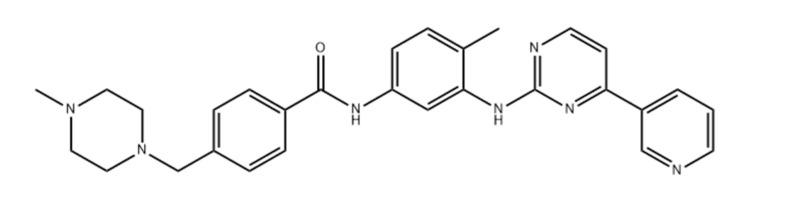
Chemical structure of imatinib.

**Figure 2 pharmaceutics-13-01086-f002:**
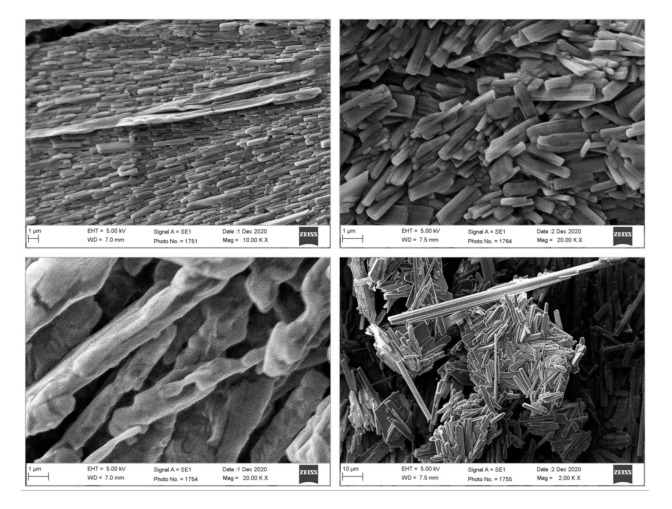
SEM images of the lyophilized NANIMAs including, STIN (**upper left**), S2TIN (**upper right**), TSIN (**lower left**), and T2SIN (**lower right**).

**Figure 3 pharmaceutics-13-01086-f003:**
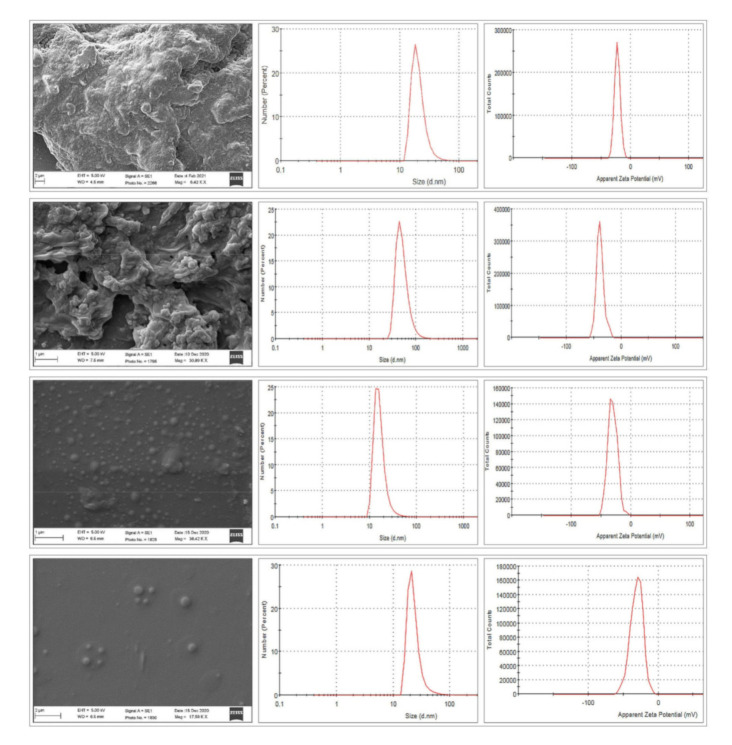
SEM (obtained by the second method), size and zeta potential of STIN (**first from top**) (size = 78.98 d.nm, ZP = −28.0 mV), S2TIN (**second from top**) (size = 94.28 d.nm, ZP = −38.2 mV), TSIN (**third from top**) (size = 88.65 d.nm, ZP = −30.5 mV), and T2SIN (**bottom**) (size = 91.11 d.nm, ZP = −34.1 mV) (*n* = 1).

**Figure 4 pharmaceutics-13-01086-f004:**
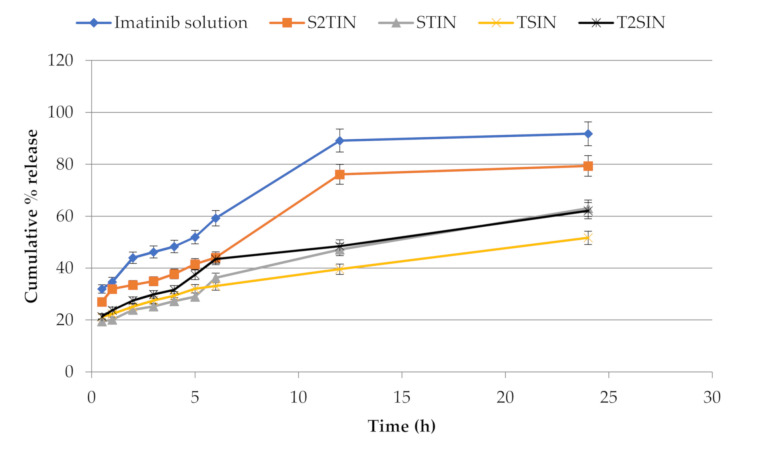
In vitro release pattern of the NANIMA formulations.

**Figure 5 pharmaceutics-13-01086-f005:**
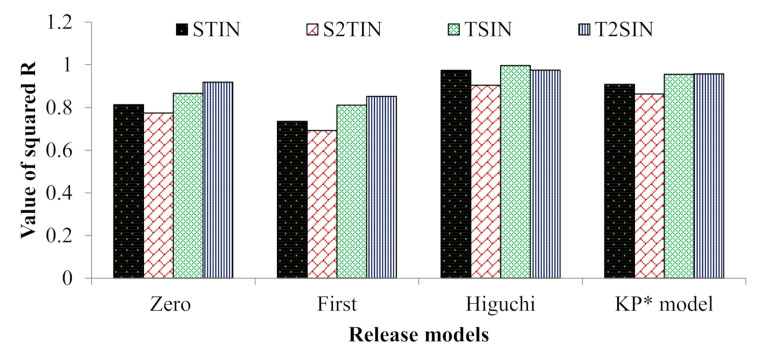
Correlation coefficient (R^2^) values for the IMA release from different NANIMA formulations (* Korsmeyer–Peppas).

**Figure 6 pharmaceutics-13-01086-f006:**
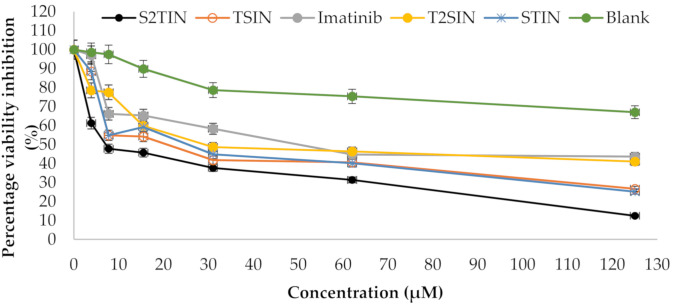
Cytotoxicity of IMA (pure) and its formulations at 72 h. Values are represented as the average of three replicates.

**Figure 7 pharmaceutics-13-01086-f007:**
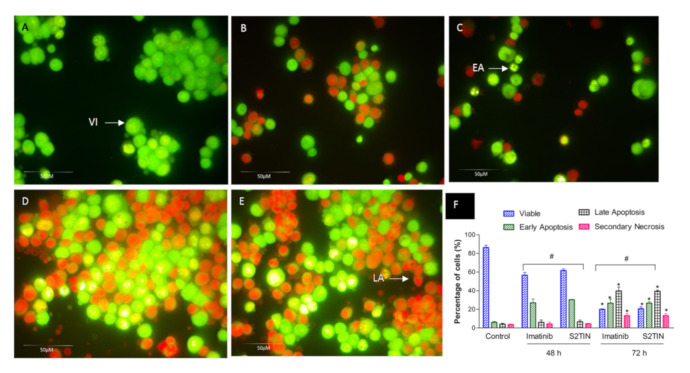
Fluorescent micrographs of acridine orange/propidium iodide double-stained MCF-7 breast cancer cells treated with IC_50_ concentrations of IMA and S2TIN for 48 and 72 h. (**A**) The untreated cells show a normal structure without prominent apoptosis or necrosis (bright green). (**B**,**C**) Early apoptosis features seen after 48 h treatment (bright green) with IMA and S2TIN, respectively. (**D**,**E**) Late apoptosis was observed at 72 h (orange color) with IMA and S2TIN, respectively. (**F**) Percentages of viable, early apoptotic, late apoptosis, and secondary necrotic cells after treatment. * *p* < 0.05 compared to the control; # no significant difference between groups. VI, viable cells; EA, early apoptosis; LA, late apoptosis. Each experiment was performed in triplicate.

**Table 1 pharmaceutics-13-01086-t001:** Amount of surfactants used in the preparation of NANIMA formulations.

Formulation Code	Sodium Lauryl Sulphate (SLS) (mg)	Tween 80 (T80) (µL)
STIN	50	25
S2TIN	75	25
TSIN	25	75
T2SIN	25	100

**Table 2 pharmaceutics-13-01086-t002:** Characterization (size, polydispersity index (PdI), zeta potential, and encapsulation efficiency (EE)) outcomes of the NANIMAs.

Formulation Code	Size (d.nm) (*n* = 3) (±SD)	PdI (*n* = 3) (±SD)	Zeta Potential (mV) (Negative) (*n* = 3) (±SD)	EE (%) (*n* = 3) (±SD)
STIN	93 ± 23	0.301 ± 0.07	24.83 ± 2.90	98.50 ± 2.43
S2TIN	105 ± 10	0.227 ± 0.06	27.70 ± 9.30	99.79 ± 0.03
TSIN	111 ± 20	0.313 ± 0.07	26.40 ± 5.09	99.63 ± 0.53
T2SIN	103 ± 12	0.312 ± 0.08	29.57 ± 3.94	99.66 ± 0.13

**Table 3 pharmaceutics-13-01086-t003:** IC_50_ value of imatinib (pure) and its formulations at 72 h.

Formulation	IC_50_ (µM)
Imatinib (pure)	52.5 ± 3.24
STIN	13.5 ± 1.0
S2TIN	6 ± 0.2
TSIN	24 ± 1.8
T2SIN	31 ± 1.9
Blank ***	≥100

*** S2TIN without imatinib.

## Data Availability

Not applicable.

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
