# Peer review of "Preparation, Characterization, and Anti-Cancer Activity of Nanostructured Lipid Carriers Containing Imatinib"

_pharmaceutics, 2021, doi:10.3390/pharmaceutics13071086_

Round 1

Reviewer 1 Report

The authors have made the corrections indicated by the reviewer. However, the figures should be improved becuse they aren't sharp enoguh.

Author Response

The figures (Figure 3, 5, 7) have been edited to increase their sharpness. Please find in attachment.

Reviewer 2 Report

Accept in present form.

Author Response

The manuscript (ID - 1272791) has been reviewed extensively; comments were raised by the expert reviewers and all the comments were responded well.  

Thanks

Reviewer 3 Report

The manuscript entitled "Preparation, characterization and anti-cancer activity of nanostructured lipid carriers containing imatinib" should be reviewed before being published in Pharmaceutics. In particular, the conclusions obtained in the physicochemical characterization of these systems are not significant, that is, the results for each of the formulations are similar within the experimental errors. The authors should strongly argue their answer.

Author Response

The manuscript (ID - 1272791) has been reviewed extensively; comments were raised by the expert reviewers and all the comments were responded well. Please check attachment.

Round 2

Reviewer 3 Report

Sorry for the inconvenience. Perhaps, my comments have been ambiguous. But the reviewer was referring to the fact that the results obtained for each system investigated are very similar. Therefore, the authors should strongly argue this issue. 

Author Response

Please find attached file containing the response of comment.

Round 3

Reviewer 3 Report

This manuscript is suitable for publication in Pharmaceutics.

This manuscript is a resubmission of an earlier submission. The following is a list of the peer review reports and author responses from that submission.

Round 1

Reviewer 1 Report

The manuscript is premature and need more work to be considered for publication. 

  • massive English editing is needed.
  • EE% is not validated and results cannot be accepted.
  • The UV methods to quantity the drug must be validated
  • The number of tables and figures should be reduce and some of the data could be moved to supplementary data. As I mentioned don my report. The manuscript is premature, needs to be extensively revised and represented. It is confusing and hard to follow with plenty of tables and figures. Authors could merge some figures and move some data to supplementary such as the method development data and the effect of temperature..

Reviewer 2 Report

Presented paper is devoted to the problem of preparation of the new therapeutic forms of cytostatic drug with high anti-cancer activity and effective delivery to the targeted organ. Lipid nanocarriers revealed nowadays a great potential for delivery of numerous anti-cancer agents. For preparation of lipid nanocarriers (NLC) the authors proposed a hot homogenization method, which is based on mixing of hot aqueous solution of surfactants (Sodium lauryl sulphate, with hot molten mixture of stearic acid and sesame oil. NLC obtained were loaded by Imatinib. This drug is effective against chronic myelogenous leukemia (CML) cells and gas-trointestinal stromal tumors. Imatinib (IMA) is a tyrosine kinase inhibitor that usually inhibits the tyrosine kinase action. The prepared nanoforms were characterized for size, polydispersity index, zeta potential, encapsulation efficiency, release studies, stability studies and MCF cells cytotoxicity studies. The optimized samples revealed average particle size of (104.63 ± 9.55) nm, PdI of 0.227 ± 0.06 and EE of 99.79 ± 0.03. All nanoformes obtained revealed slow and sustained release behavior. The surfactants used in the preparation of NLC effected on particle size, zeta potential, encapsulation efficiency, stability and Ima-release. It was shown the cytotoxicity in 8.75 times increase for cytotoxicity of optimized IMA-loaded NCL (IC50 = 6 µM) when compared with IMA alone (IC50 = 52.5 µM) against MCF-7 breast cancer cells. After this, NLC containing IMA possibly will be employed for the cure of breast cancer. A smaller amount of IMA loaded NLC will be sufficient than IMA alone for the treatment of breast cancer combined with decreasing of adverse effects as in case of IMA alone, and proposed nanoforms IMA-loaded NCL could be useful for anti-cancer therapy.

   Thus, the paper consists of new and important results showing the effective therapeutic outcome of prepared nanoforms of Imatinib in breast cancer treatment and their possible use in pharmaceutical practice.the paper consists of new and important results showing the effective therapeutic outcome of prepared nanoforms of Imatinib in breast cancer treatment and their possible use in pharmaceutical practice. Presented paper can be published after minor revisions and English editing

Please, also check the details in Experimental Part, for example, “Scanning electron microscope (Zeiss EVO LS10; United Kingdom)” – possibly Zeiss is belonging to Germany

Reviewer 3 Report

The authors present an interesting work aimed to the optimization of a nanostructured lipid carrier for imatinib. The study covers several aspects involved in the synthesis, characterization and cytotoxicity of the nanosystems. However, despite of the interest of the subject, the manuscript lacks the required quality for publication in Pharmaceutics. The contents are mainly imprecise and instead of defined values, the word certain is used too often for instance in line 220: “… furthermore, a certain value of ZP….”

Section 3.1 is too long, shoud be summarized and rewritten with more accurate interpretation of the concepts. PDI gives information not only about aggregates in the sample but also about the sie of the particles case. A sample can contain big particles but not agglomerated. The zeta potential section is too long and imprecise. See lines 228-231. From lines 235, at what refers to: “The lower ZP value can be explained….”What is the meaning of the sentence of lines 244 and 245?

The legend of figure 5 states: Effect of temperature on particle size of NANIMAs after 3 months of storage. An increase in particle size was observed to be non-significant in all NANIMA formulations. But there is no data about temperature and there is a visible change in size with time. The same applies for figure 6.

Zeta potential in figure 8 is clearaly negative but the values in table 2 are positive.

Table 3. What is the blank sample?In fact all the fomulations without drug should have been tested.

Reviewer 4 Report

Manuscript ID: pharmaceutics-1227649

the manuscript of Hafiz Antar Makeen, Syam Mohan, Mohamed Ahmed Alkasim, Muhammad Hadi Sultan, Ahmed A Albarraq, Rayyan A Ahmed, Hassan A. Alhazmi, Mohammad Intakhab Alam as Co-authors: “Preparation, characterization and anti-cancer activity of nanostructured lipid carriers containing imatinib” presented actual studies regarding the development of new nanostructurated lipid carriers for delivery of imatinib, their preparation and characterisation. Topic seems to be suitable for researchers who worked in related areas. However, there are some important issues needed to be addressed before publication.

Major:

  1. Paragraph – Materials and Methods
    • subparagraph 2.1.:

- please indicate the temperature of the aqueous solution;

-justify chosen ratio of SLS and T80 in compositions.

  • subparagraph 2.2.:

-authors wrote, that “The study was performed after proper dilution of formulations with Millipore water” Please, indicate the final concentration of components or indicate of dilution level.

  • subparagraph 2.3.:

-justify choice of wavelength 265 nm for the measurement.

  1. Paragraph – Results and discussions.
    • Please give a more detailed description of the procedure for preparation of NANIMA – justify the choice of all used components - surfactants, lipid phase and their ratio.
    • Please reduce the information in subparagraph 3.1. The authors made a wide discussion, but obtained data for the size of particles and their PDI values are similar. Taking into account the given ±SD ranges the particle sizes for all compositions are very close.
    • Table 2. There are do not need to give decimal numbers for describing particle sizes.
    • Subparagraph 2.3. The need for two samples is not defined. Please identify which data from table 2 is detected for which sample?
    • Figure 2 and its legend. Data is different as described in Table 2. Please, give any comparison and explanation. Also, an explanation of zeta-potential values is required. In Table 2 all ZP values are positive, Fig.2. – three negative and only for T2SIN is positive ZP value, while in Fig. 8 all ZP values are negative. Please explain it.
    • Subparagraph 3.5. I am not sure, that authors use the right terminology. In the beginning, the authors underlined that “Among the factors affecting the stability of lipid nanocarriers including NLCs, the temperature is considered to be a significant one. Storage at room temperature (25°C) does not require any special condition. Thus, stability at room temperature (25°C) is considered to be commendable.” I suggest, that it mean that all experiments were performed at constant temperature - room temperature (25°C). However, all legends of Figures 5-8 contain introduction – effect of temperature on.... It seems, that more appropriate will be term – effect of storage...
    • Please justify the period of chosen storage time – 3 month. Why 3 month? What happens after 1 month or 5-6 months?
    • Figure 5. Authors suggested – “An increase in particle size was observed to be non-significant in all NANIMA formulations.” However, mainly for all cases, the increment in particle size is near 50%, from 100 to 150...
    • . Subparagraph 3.6:

- Figure 9. Please change the image of the mentioned figure. The existing figure is unclear.

-Authors performed cytotoxicity studies of obtained formulations on MCF-7 breast cancer cells. However, additional data regarding cytotoxicity against normal (non-cancer) cells of compositions also is necessary to provide for the evaluation of composition properties.

-Table 3. Studies of cytotoxicity of all pure components of existing formulations would be recommendable.

  1. In conclusions – give more detailed data regarding your results in the context of general information.

Minor:

  1. Structure of imatinib should be added in the manuscript.
  2. In line 56, please remove space before .., and...
  3. Line 427, please use subscript for 50 in the term IC50.

Consequently, I do recommend accepting this manuscript for publication with major revision.

Round 2

Reviewer 3 Report

The authors have improved the manuscript by addressing the questions of the reviewers. However, despite their efforts, there are still some points that make the manuscript faulty.

Major points

  1. Is it correct to talk about entrapment efficiency after 3 months? Is it not more appropriate to refer to leakage upon time, parameter that gives an idea about sample stability?
  2. As mentioned before, polydispersity of lipid carriers, except for sample S2TIN is too high for them to be considered as drug delivery systems. Thus, the starting systems lack of the required properties for further experiments.
  3. Zeta potential becomes less negative upon storage. The authors attribute it to aggregation. However, such negative potential would guarantee physical stability. Could it be due to the variety in particle sizes as indicated by the PDI?

Minor points.

  1. If possible, colors in figure 4 would enrich its quality.
  2. Fig 10. X-axis should be changed to concentration.

Reviewer 4 Report

The main suggestions of the reviewer are included in the manuscript after rework.

However, there are some additional issues needed to be addressed before publication:

  1. Please add at least briefly some literature data regarding cytotoxicity of all pure components of the tested formulations if there is no experimental data.
  2. From the previous report. Comment 4-5: Please give a more detailed description of the procedure for preparation of NANIMA – justify the choice of all used components - surfactants, lipid phase and their ratio.

Author answer 4-5: It was corrected and added in the manuscript. The following statements were added in the manuscript - “The preliminary studies were conducted and the different ingredients including surfactants, lipid phase and their ratio were selected based on solubility, stability, availability and particle size.”

Please justify this general sentence, add provide more detailed information. If it is possible - support your assumptions with the relevant literature data.

  1. Please carefully check the style point of the manuscript, use space between word figure and number for figures 9 and 10.

Consequently, I do recommend accepting this manuscript for publication with minor revision.